# Arab Countries and Oncology Clinical Trials: A Bibliometric Analysis

**DOI:** 10.3390/cancers15184428

**Published:** 2023-09-05

**Authors:** Humaid O. Al-Shamsi, Ibrahim Abu-Gheida, Kareem Sameh, Nouran E. Tahoun, Khaled M. Musallam

**Affiliations:** 1Burjeel Cancer Institute, Burjeel Medical City, Abu Dhabi P.O. Box 92510, United Arab Emirates; ibrahim.abugheida@burjeelmedicalcity.com (I.A.-G.); khaled.musallam@burjeelholdings.com (K.M.M.); 2College of Medicine, Gulf Medical University, Ajman P.O. Box 4184, United Arab Emirates; 3Emirates Oncology Society, Dubai P.O. Box 6600, United Arab Emirates; 4College of Medicine, University of Sharjah, Sharjah P.O. Box 27272, United Arab Emirates; 5Gulf Cancer Society, Alsafa P.O. Box 2311, Kuwait; 6Pfizer Gulf FZ LLC, Dubai Media City, Dubai P.O. Box 502749, United Arab Emirates; kareem.sameh@pfizer.com (K.S.); nourane.e.tahoun@pfizer.com (N.E.T.)

**Keywords:** research, oncology, Arab, Middle East, North Africa, cancer management

## Abstract

**Simple Summary:**

The burden of cancer is growing in Arab countries, which have different patient characteristics, cancer profiles, and cancer care capacities compared with other regions. Therefore, there is a need for cancer trials that focus on people in the Arab region. This study looked at the records of cancer trials with a participating Arab center published between 2000 and 2021. Overall, Arab countries participated in 320 published clinical trials related to cancer treatment. The number of trials, number of patients included, and the number of trials with more than one participating site increased over time. Trials with participating Arab and non-Arab countries included more patients and were more likely to receive funding from external sources than Arab-only trials. Joint efforts are needed to overcome barriers to cancer trial participation and encourage international support for cancer research in the Arab region.

**Abstract:**

The increasing cancer burden is a major health concern in Arab countries with cross-regional variations in cancer profiles. Given the limited oncology research output and scarce data on cancer trial participation in the Arab region, this study explored the therapeutic cancer trial landscape in Arab countries over the past 20 years. A bibliometric analysis of the PubMed database was conducted on primary publications of therapeutic trials with a participating Arab center. Arab countries participated in 320 published cancer-related therapeutic trials (2000–2021). During this period, there was a consistent increase in the number of trials, sample size, multiregional site participation, and number of randomized trials. However, most trials were small, did not receive external funding, and included a single Arab site. Compared with Arab-only trials, trials with joint non-Arab sites were larger (*p* = 0.003) and more likely to be externally funded (*p* < 0.001). Citation numbers and journal impact factors were higher in trial publications with joint non-Arab authorship than those without (*p* < 0.001, for both). Despite improving conduct and publication records of oncology trials with Arab centers, cancer trial participation remains limited in Arab countries. Concerted efforts are required to encourage sponsorship and international collaboration in this region.

## 1. Introduction

Arab countries are commonly defined as the 22 Arabic-speaking member states of the League of Arab States across the Middle East and North Africa, with a total population of 444,517,783 in 2021 [1].

The increasing cancer burden is a major health concern in Arab countries. Between 2018 and 2020, the estimated number of new cancer cases in Arab countries increased from 429,325 to 463,675, accounting for 2.4% of the global cancer incidence [2]. In 2020, breast, colorectal, and liver cancers ranked the highest in incidence among Arab women, whereas lung, liver, and prostate cancers had the highest incidence in Arab men [2,3,4].

Cancer is also a major cause of mortality in Arab countries, causing an estimated 281,656 total deaths in 2020, corresponding to 2.8% of global cancer deaths [3]. Notably, the cancer mortality-to-incidence ratio (MIR) in Arab countries is higher than the global rates for both men (0.68 vs. 0.55) and women (0.54 vs. 0.48) [2,3]. In 2020, Yemen had the highest MIR for both men (0.79) and women (0.69) among Arab countries, whereas the lowest MIR was reported in the United Arab Emirates (UAE) for both men (0.33) and women (0.47) [2,3].

The ethno-linguistic, sociocultural, and historical links between Arab countries may indicate genetic and lifestyle similarities among Arab populations [5,6]. According to the World Health Organization (WHO) 2020 report on regional cancer profiles, tobacco smoking, infections, obesity, and exposure to ultraviolet light were the common contributors to cancer cases or deaths in the Eastern Mediterranean Region (EMR), which includes 19 Arab countries (except Algeria, Comoros, and Mauritania) [7]. However, Arab states demonstrate considerable differences in demography, socioeconomic and political status, population health and public health policies, healthcare services, and clinical research ecosystems [4,8,9,10] that can explain the country-level variations in cancer incidence and mortality among Arab men and women [4,11,12]. For instance, there is a clear disparity in cancer burden between high- and low-income Arab countries. Colorectal, prostate, and lung cancers were the top three cancers in men from high-income Arab countries in 2018. In contrast, stomach, esophageal, and lung cancer were the cancers with the highest incidence in men from low-income Arab countries [13]. Among women, breast and uterus cancers had the highest incidence in high- and low-income countries, respectively, whereas thyroid and esophageal cancers ranked third in incidence in high- versus low-income Arab countries [13].

Despite the growing cancer burden and the differential drivers and characteristics of various cancers within the region, oncology research output in Arab countries is low [11,14,15,16]. Overall, Arab institutions published 26,656 cancer-related studies from 2005 to 2019, contributing to 1.5% of global cancer publications during this period. Of note, >50% of cancer-related literature in the Arab region came from two countries, Egypt and Saudi Arabia, and most of the published articles were related to breast (*n* = 2241), colorectal (*n* = 1169), or liver (*n* = 1017) cancers [16]. Additionally, there are numerous limitations and variations in cancer screening, cancer registries, patient access, and oncology care systems in Arab countries. For example, high-quality national cancer registries were available in only 28% of EMR countries, and no country had high-quality mortality registration. In addition, the functionality of cancer control and oncology care systems in the EMR are subject to some limitations in infrastructure, workforce, and management capacity [7].

Region-specific and country-level differences in genetic predisposition, environmental exposures, socioeconomic status, cancer treatment availability and access, and pharmacogenomics demand locally relevant clinical trials in Arab populations [7,17]. However, it is well noted that there are limited patient-level data and cancer trial participation in this region [18]. This is a critical gap as the findings of clinical trials in Western populations are not necessarily applicable to the populations in Arab countries, posing challenges to cancer prevention, screening, diagnosis, treatment, and survivorship in the Arab states [19]. Therefore, it is important to systematically assess the available evidence on oncology trials in Arab countries to identify the existing gaps and define key priorities in this area.

The present study aims to explore the therapeutic cancer trial landscape in the Arab region over the past 20 years, with a focus on interventional oncology trials involving pharmacotherapy, radiation therapy, surgical therapy, behavioral/psychological therapy, herbal/alternative therapy, and other therapeutic interventions. In this work, we relied on literature sources to identify published therapeutic cancer clinical trials involving centers from Arab countries, and analyzed relevant characteristics and trends related to the trials and publication of their results to recognize persisting unmet needs that should drive future action. From the identified trials, we examined both the publication-related parameters (e.g., publication authorship, journal impact factor, and number of citations per publication) and the trial-related characteristics (e.g., number of Arab country sites, number of participating sites from non-Arab countries, sample size, randomized trial design, target cancer, therapeutic intervention type, and funding source).

## 2. Materials and Methods

For this work, we considered Arab countries as defined by the World Bank, i.e., Algeria, Bahrain, Comoros, Djibouti, Egypt, Iraq, Jordan, Kuwait, Lebanon, Libya, Mauritania, Morocco, Oman, Palestinian Territories, Qatar, Saudi Arabia, Somalia, Sudan, Syria, Tunisia, UAE, and Yemen [1].

### 2.1. Literature Search

We first searched for English-only journal publications using PubMed on December 31, 2021, using the following search terms: (cancer*[tiab] OR tumour*[tiab] OR tumor*[tiab] OR neoplas*[tiab] OR malignan*[tiab] OR carcinom*[tiab]) AND (“middle East” OR “middle eastern” OR “Gulf states” OR “gulf state” OR Bahrain* OR Iraq* OR Kuwait* OR Oman* OR Qatar* OR “Saudi Arabia” OR “saudi arabian” OR “United Arab Emirates” OR UAE OR Emirati OR Algeria* OR Egypt* OR Jordan* OR Lebanon OR Lebanese OR Morocc* OR Tunisia* OR “north africa” OR “north african” OR “Comoros” OR Djibouti* OR Libya* OR Mauritania* OR Somalia* OR Palestine OR Palestinian* OR Sudan OR Sudanese OR Syria* OR Yemen* OR “Middle East”[Mesh] OR “Bahrain”[Mesh] OR “Iraq”[Mesh] OR “Kuwait”[Mesh] OR “Oman”[Mesh] OR “Qatar”[Mesh] OR “Saudi Arabia”[Mesh] OR “United Arab Emirates”[Mesh] OR “Algeria”[Mesh] OR “Egypt”[Mesh] OR “Jordan”[Mesh] OR “Lebanon”[Mesh] OR “Morocco”[Mesh] OR “Tunisia”[Mesh] OR “Africa, Northern”[Mesh] OR “Comoros”[Mesh] OR “Djibouti”[Mesh] OR “Libya”[Mesh] OR “Mauritania”[Mesh] OR “Somalia”[Mesh] OR “Sudan”[Mesh] OR “Syria”[Mesh] OR “Yemen”[Mesh]). The search was restricted to “Clinical Trial” and to the years 2000 to 2021 to reflect a contemporary analysis. This retrieved 1428 journal publications that were reviewed by three authors (title + abstract and full-text publication when necessary) to exclude: (1) publications that did not report data from therapeutic cancer clinical trials; (2) secondary publications of clinical trials; and (3) publications of clinical trials that did not include a center from an Arab country for recruitment (even if the publication included an author from an Arab country). The latter was identified through the methods section, other in-text mentions or sources, ethical approvals, informative authorship and contributions, protocol supplement, or trial registry. For each journal publication of a clinical trial, we retrieved the following variables: year of publication; journal impact factor during the year of publication where available (retrieved through Journal Citation Reports^TM^, Clarivate, St. Helier, Jersey); number of citations (retrieved on 29 March 2022 through PubMed and Europe PubMed Central); Arab country authorship (no authors affiliated with Arab countries, authors(s) affiliated with a single Arab country, or author(s) affiliated with multiple Arab countries); non-Arab country authorship (no authors affiliated with non-Arab countries, or authors affiliated with Europe, North America, Central or South America, Asia/Australia, Africa, or multiple regions); first author(s) (affiliated with Arab country or not); and corresponding author(s) (affiliated with Arab country or not). We also retrieved the following information about the clinical trial: sample size; Arab country site(s) (single site from a single country, multiple sites from a single country, or multiple sites from multiple countries); non-Arab country site(s) (none, Europe, North America, Central or South America, Asia/Australia, Africa, or multiple regions), cancer type (breast, gynecologic, colorectal, upper gastrointestinal/liver/pancreas, lung, head and neck/thyroid, prostate, kidney/bladder, leukemia/lymphoma/myeloma, brain, soft tissue/bone, skin/melanoma, multiple, other/non-specific); intervention type (pharmacologic therapy, radiation therapy, surgical therapy, behavioral/psychological therapy, herbal/alternative therapy, other therapy, or multiple therapy); randomized controlled trial design (yes or no); and external funding (none/undisclosed, pharmaceutical, non-pharmaceutical, or both).

### 2.2. Statistical Analysis

Descriptive statistics were presented as median (range) or percentages. Bivariate comparisons were made using the chi-squared test for categorical variables and Mann–Whitney U Test for continuous variables. All *p*-values were two-sided with the level of significance set at <0.05.

## 3. Results

A total of 320 therapeutic cancer clinical trial publications that contained evidence of the participation of center(s) from Arab countries were included in this analysis. All included publications were published between the years 2000 and 2021, with an evident increase in the number of publications over time: 2000–2006 (*n* = 44, 13.8%), 2007–2013 (*n* = 105, 32.8%), and 2014–2021 (*n* = 171, 53.4%).

### 3.1. Publication-Related Analyses

A summary of authorship for journal publications of a therapeutic clinical trial that included the participation of a center from one of the included 22 Arab countries is provided in Table 1. Most (91.3%) publications included author(s) from a single Arab country, with a slight increase in authorship from multiple Arab countries in recent years. In the majority of publications, Arab authors were listed as the first author (92.2%) or the corresponding author (89.7%), with a decreasing trend in recent years (2014–2021). Joint authorship with non-Arab countries was consistently low over the period analyzed (21.3%) and primarily included authors from Europe and North America. Notably, a higher proportion of publications included multiregional rather than single-country authorship in recent years. There was no significant difference in the frequency of joint non-Arab authorship between single or multiple Arab country authorship publications (20.9% vs. 25%, *p* = 0.61).

A total of 270 (84.4%) trials were published in journals with a Journal Citation Reports impact factor (36/44 [81.8%] in 2000–2006, 80/105 [76.2%] in 2007–2013, and 154/171 [90.1%] in 2014–2021). The overall median impact factor was 2.57 (range, 0.08–35.9) and increased from 2000 to 2021: 1.87 (range, 0.34–11.12) in 2000–2006, 2.04 (range, 0.08–16.95) in 2007–2013, and 2.90 (range, 0.77–35.92) in 2014–2021. The journal impact factor was significantly higher in publications with joint non-Arab authorship (median, 3.3; range, 0.49–35.9) than those without (median, 2.49; range, 0.08–10.1; *p* < 0.001). The overall median number of citations per article was 4 (range, 0–223) and was lower for more recent years: 5.5 (range, 1–139) in 2000–2006, 4 (range, 0–71) in 2007–2013, and 4 (range, 0–223) in 2014–2021. The number of citations was also significantly higher in publications with joint non-Arab authorship (median, 7; range, 0–223) than those without (median, 4; range, 0–139; *p* < 0.001).

### 3.2. Clinical Trial-Related Analyses

Characteristics of the clinical trials featured in the journal publications are summarized in Table 2. The number of trials conducted by individual Arab countries is illustrated in Figure 1. There was a cumulative increase in the median sample size of trials and a substantial increase in the proportion of randomized controlled trials over time (36.4% in 2000–2006, 46.7% in 2007–2013, and 75.4% in 2014–2021). Breast (25%), upper gastrointestinal/liver/pancreas (12.2%), and colorectal (8.1%) cancers were consistently the leading cancer types in the included clinical trial publications (Table 2). Pharmacologic (50%), surgical (14.7%), and radiation (5.3%) therapy were the leading interventions in included clinical trial publications across the observation period, with multiple therapies used in 18.1% of publications. Most trials included only a single site from a single Arab country (82.8%) and did not receive external funding from pharmaceutical companies or other sources (82.8%). Overall, joint participation of non-Arab sites was consistently low over time (12.5%); however, the proportion of trials with multiregional site participation increased over the years. The proportion of randomized controlled trials was comparable between trials with joint non-Arab country sites and those without (62.5% vs. 60.4%; *p* = 0.80). However, external funding was significantly more common in trials with joint non-Arab country sites than in those without (72.5% vs. 9.3%; *p* < 0.001). The trial sample size was also significantly higher in trials with collaborating sites in non-Arab countries (median, 96; range, 9–2577) than in those without (median, 50; range, 20–657; *p* = 0.003).

## 4. Discussion

This study provides several insights on the participation of Arab countries in therapeutic cancer clinical trials over the last 20 years. Overall, there was a steady increase in the number of trials conducted between the years 2000 and 2021. Although the clinical trials included a variety of interventions, they primarily reported data from trials in a few common cancers such as breast and gastrointestinal cancers. Most of the included publications were of small clinical trials conducted by single sites within a few individual countries. Although a considerable proportion of the published clinical trials were randomized clinical trials, the availability of external funding was low. These factors could explain the low impact factor for the majority of published therapeutic cancer trials from Arab countries. Previous research has indicated that author nationality, number and distribution of collaborating centers, sample size, and funding status of cancer trials are independent predictors of their publication in medical journals [20,21,22]. Tang et al. [20] showed that phase 3 oncology trials with larger sample size (odds ratio [OR] 1.08, 95% confidence interval [95% CI] 1.00–1.15) and external sponsorship (OR 1.69, 95% CI 1.04–2.74) were more likely to be published in high-impact-factor cancer-focused journals relative to the trials with a smaller sample size and no external funding. International authors from non-European or European countries were also found to be less likely to publish in high-impact-factor journals than those from North America (adjusted OR 0.17, 95% CI0.06–0.48; and adjusted OR 0.13, 95% CI 0.06–0.29, respectively). Furthermore, Van den Bogert et al. [21] reported that international multicenter trials conducted either within the European Union (adjusted OR 2.2, 95% CI 1.1–4.4) or outside (adjusted OR 2.0, 95% CI 1.0–4.0) had a higher likelihood of publication in peer-reviewed journals compared with the single-center trials.

These prior observations and predictors were largely aligned with the findings of the current study. For trials conducted in collaboration with non-Arab countries, the sample size was larger and external funding was more common than for trials conducted in Arab countries only. These observations indicate an expanding pool of investigators in Arab countries, but also suggest some degree of reliance on Western countries for inclusion in large, externally funded studies. Moreover, the frequency of collaboration between Arab and non-Arab countries remains low, despite marginal improvement in recent years.

These findings are also reflected in the associated journal publications, with “better” publication outcomes, i.e., a higher journal impact factor and article citations, noted when Arab countries are involved in trials in collaboration with non-Arab countries compared with conducting them alone.

The total number of publications (*N* = 320) for cancer-related therapeutic trials hosted by or conducted in collaboration with Arab countries between 2000 and 2021 indicates a low research output for a region constituting 5.6% of the world population and contributing to 5% of global growth domestic product (GDP) in purchasing power standards [23,24]. Of note, most published therapeutic cancer trials in the Arab region (73%) had a participating site from Egypt, corroborating previous findings by Machaalani et al. [16] who showed that Egypt contributed to the largest number of cancer-related publications in Arab countries between 2005 and 2019, both in total and per national GDP. Egypt is the most populous Arab state [23], with the highest cancer incidence, 5-year prevalence, and mortality in 2020 [3]. Among Arab countries, Egypt and Qatar have the largest proportion of cancer research institutions ranked among the top 30 institutions from the Middle East. Furthermore, Egypt has the highest number of full-time equivalent health researchers per million persons in the EMR [25,26]. A detailed comparison of the publication outcomes and trial-related parameters between Egypt and other Arab countries in a future study could further elucidate the reasons for the observed disparity.

Overall, therapeutic oncology trial publications from Arab countries were proportionally small relative to total cancer trial publications indexed in PubMed between 2000 and 2021 (320/110,324), increasing gradually but remaining consistently low during this period (44/27,927 in 2000–2006, 105/35,188 in 2007–2013, and 171/48,887 in 2014–2021). The observed publication output from Arab countries is very limited in the context of total oncology trials (both observational and interventional) listed in the WHO International Clinical Trials Registry Platform for the period 1999–2021 (*N* = 104,491) and all interventional cancer trials registered at ClinicalTrials.gov by December 2022 (*N* = 75,629) [27]. This is in line with the trends for the number of clinical trials by WHO regions for the period of 1999–2021, where the EMR (*n* = 5423) was behind the following regions: Western Pacific (*n* = 16,860), Europe (*n* = 14,879), Americas (*n* = 12,969), and most recently, Southeast Asia (*n* = 9374) [27].

In addition, there is some inconsistency between the observed proportion of cancers targeted in the therapeutic trials conducted in Arab countries and those studies in oncology trials led by high- or middle-income countries. While breast and gastrointestinal cancers were the cancers most commonly included in the published articles in this analysis, hematologic, breast, gastrointestinal, lung, and urologic cancers were the leading targets in global phase 3 therapeutic trials published between 2014 and 2017 [28].

The focus of Arab clinical trials on breast and gastrointestinal, including colorectal and liver, cancer aligns with the high incidence and mortality of these cancers in the Arab population [2,3]. However, the number of clinical trials on lung, thyroid, gynecologic, and urologic cancers is disproportionately low despite their substantial burden in Arab men and women [2,3,13,29].

Furthermore, Arab countries had a lower share of gross domestic expenditure on research and development per GDP compared with the global research expenditure (0.64 vs. 1.93) in 2020, lagging behind Central and Eastern Europe (1.10), East Asia and the Pacific (2.29), and North America and Western Europe (2.89) [30]. Between 2005 and 2019, each Arab country (excluding Egypt) published <50 cancer-related articles per billion USD of their GDP, with the lowest contribution from low-income countries [16]. Notably, the participation of middle-income Arab countries (Algeria, Egypt, Lebanon, Morocco, and Tunisia) in phase 3 oncology trials and their corresponding bibliometric research output were proportionally lower than non-Arab middle-income countries such as China, India, and Turkey [28]. The lower expenditure on research translates into the lower cancer research output and clinical trial participation in Arab countries relative to the global trend.

These findings highlight a considerable opportunity across the wider Arab region for participation in, and publication of, therapeutic oncology trials. However, there are multiple barriers to clinical cancer research in Arab countries, including: a limited training and engagement of healthcare professionals, poor institutional collaborative research, a lack of clinical trial units and specifically designed system operating procedures for research, inconsistent regulatory processes for novel therapies in some countries, challenges surrounding patient enrollment in cancer trials, underrepresentation in, or restricted access to, global cancer-centric research and knowledge dissemination, and insufficient support or funding from the governments and pharmaceutical industry [18,28,29,31,32,33].

Overcoming the existing barriers in clinical cancer research and oncology care requires concerted multi-tiered actions across the Arab region. The lack of high-quality data from patients with cancer in Arab countries warrants the optimization of national cancer registries and improvements in data access [34,35]. The development of designated oncology research centers in Arab countries could improve the recruitment and retention of skilled researchers and clinicians, who would otherwise seek new opportunities elsewhere [33,36]. Furthermore, the incorporation of research skills in the national medical education system, postgraduate training, and clinical practice can increase healthcare professional engagement in research and evidence generation [18,33,37]. On an institutional level, initiatives to encourage multidisciplinary intra-country and inter-country collaboration between the public and private sectors could strengthen local and international research networking, expanding clinical trial capacity and access in the region and enabling multicenter and potentially multinational oncology trials [18,35]. Additionally, public education and awareness plans could change misperceptions and trust issues about clinical trials among patients and the wider community, potentially improving clinical trial acceptance and accrual [18,36]. Finally, clinical trial conduct in Arab countries could be further enhanced by the standardization of research review and approval processes [38]. These advancements would address the global oncology community and pharma industry’s concerns over clinical cancer research capabilities in Arab countries, enabling efforts to include Arab institutions in pharma-sponsored clinical trials, and facilitating funding and support for locally led cancer trials in the region. Concerted actions to establish a regional oncology research center and enhance cross-institutional collaborations with leading international cancer research groups such as the European Organization for Research and Treatment of Cancer, NRG oncology, and Trans Tasman Radiation Oncology Group could help to facilitate improvements in the design, conduct, and dissemination of oncology clinical trials. Central to this collaborative engagement is the development of a harmonized cancer research framework by the regulatory bodies across the Arab region, and the establishment of cancer research units within Arab institutions to facilitate the research-led training of cancer practitioners and evidence-informed oncology care in Arab countries.

This study does not come without limitations. We relied on English-language journal publications in PubMed to identify clinical trials, which would exclude trials that were not published in a PubMed-indexed journal (results never published, results published in non-indexed journals, results published as congress abstracts only, and ongoing trials without results) and those published in non-English language. However, retrieving publications through an additional Google scholar search was not feasible or practical considering the broad nature of the topic investigated. Moreover, relying on information from congress abstracts or ClinicalTrials.gov listings would not have allowed for the retrieval of all variables of interest. In addition, some trials had a small sample size (*n* < 10) and their inclusion in the analyses might have affected the overall findings of our study. Lastly, there is a chance that some trials were not retrieved through our search as no reference to Arab countries was made in their keyword indices.

## 5. Conclusions

Despite the rapidly increasing burden of cancer among Arab populations, cancer research productivity and clinical trial capacity are still low in Arab countries. The number of cancer trials published in high-impact journals and the proportion of randomized controlled trials led by, or enrolling patients from, Arab countries is increasing. However, the lack of external funding and limited international collaboration are important limiting factors for clinical trial conduct and publication outcomes in Arab countries. These shortcomings can be addressed by developing a research-enabling oncology care system, establishing regional cancer-specific research groups, and enhancing cancer data utilization, with a subsequent beneficial impact on cancer care and patient outcomes.

## Figures and Tables

**Figure 1 cancers-15-04428-f001:**
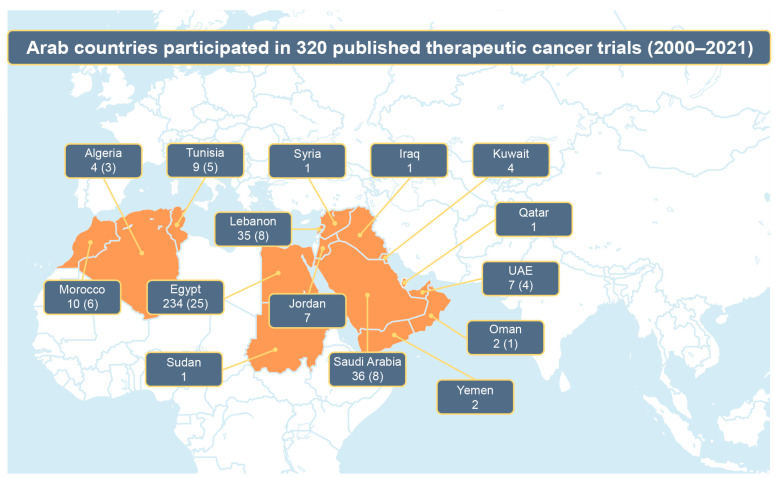
The number of therapeutic cancer clinical trials for which individual Arab countries provided participating site(s). Numbers are presented as total number of trials, with number of trials in collaboration with non-Arab countries in parentheses.

**Table 1 cancers-15-04428-t001:** Authorship of therapeutic cancer clinical trial journal publications that contained evidence of participation of center(s) from Arab countries.

Parameter, *n* (%)	Overall (*N* = 320)	2000–2006 (*n* = 44)	2007–2013 (*n* = 105)	2014–2021 (*n* = 171)
**Arab country authorship**				
None	0 (0)	0 (0)	0 (0)	0 (0)
Single country	292 (91.3)	42 (95.5)	100 (95.2)	150 (87.7)
Multiple countries	28 (8.8)	2 (4.5)	5 (4.8)	21 (12.3)
**Non-Arab country authorship**				
None	252 (78.8)	33 (75)	94 (89.5)	125 (73.1)
Any	68 (21.3)	11 (25)	11 (10.5)	46 (26.9)
Europe	29 (9.1)	10 (22.7)	5 (4.8)	14 (8.2)
North America	14 (4.4)	0 (0)	4 (3.8)	10 (5.8)
Central or South America	0 (0)	0 (0)	0 (0)	0 (0)
Asia/Australia	1 (0.3)	0 (0)	1 (1)	0 (0)
Africa	1 (0.3)	0 (0)	1 (1)	0 (0)
Multiple regions	23 (7.2)	1 (2.3)	0 (0)	22 (12.9)
**First author from an Arab country**	295 (92.2)	44 (100)	105 (100)	146 (85.4)
**Corresponding author from an Arab country**	287 (89.7)	43 (97.7)	103 (98.1)	141 (82.5)

**Table 2 cancers-15-04428-t002:** Characteristics of therapeutic cancer clinical trials with evidence of participation of center(s) from Arab countries.

Parameter, *n* (%)	Overall (*N* = 320)	2000–2006 (*n* = 44)	2007–2013 (*n* = 105)	2014–2021 (*n* = 171)
**Arab country site(s)**				
Single site from a single country	265 (82.8)	37 (84.1)	85 (81)	143 (83.6)
Multiple sites from a single country	42 (13.1)	6 (13.6)	19 (18.1)	17 (9.9)
Multiple sites from multiple countries	13 (4.1)	1 (2.3)	1 (1)	11 (6.4)
**Non-Arab country site(s)**				
None	280 (87.5)	40 (90.9)	100 (95.2)	140 (81.9)
Any	40 (12.5)	4 (9.1)	5 (4.8)	31 (18.1)
Europe	9 (2.8)	2 (4.5)	3 (2.9)	4 (2.3)
North America	5 (1.6)	0 (0)	0 (0)	5 (2.9)
Central or South America	0 (0)	0 (0)	0 (0)	0 (0)
Asia/Australia	2 (0.6)	0 (0)	0 (0)	2 (1.2)
Africa	1 (0.3)	0 (0)	1 (1)	0 (0)
Multiple regions	23 (7.2)	2 (4.5)	1 (1)	20 (11.7)
**Sample size, median (range)**	52 (2–2577)	38 (2–533)	44 (5–657)	64 (9–2577)
**Randomized, controlled trial**	194 (60.6)	16 (36.4)	49 (46.7)	129 (75.4)
**Cancer type**				
Breast	80 (25)	14 (31.8)	27 (25.7)	39 (22.8)
Gynecologic	12 (3.8)	0 (0)	9 (8.6)	3 (1.8)
Colorectal	26 (8.1)	2 (4.5)	10 (9.5)	14 (8.2)
Upper gastrointestinal/liver/pancreas	39 (12.2)	5 (11.4)	14 (13.3)	20 (11.7)
Lung	18 (5.6)	1 (2.3)	3 (2.9)	14 (8.2)
Head and neck/thyroid	10 (3.1)	1 (2.3)	3 (2.9)	6 (3.5)
Prostate	7 (2.2)	1 (2.3)	3 (2.9)	3 (1.8)
Kidney/bladder	15 (4.7)	1 (2.3)	6 (5.7)	8 (4.7)
Leukemia/lymphoma/myeloma	17 (5.3)	3 (6.8)	4 (3.8)	10 (5.8)
Brain	11 (3.4)	2 (4.5)	3 (2.9)	6 (3.5)
Soft tissue/bone	6 (1.9)	0 (0)	2 (1.9)	4 (2.3)
Skin/melanoma	4 (1.3)	1 (2.3)	0 (0)	3 (1.8)
Multiple	13 (4.1)	1 (2.3)	2 (1.9)	10 (5.8)
Other/non-specific	62 (19.4)	12 (27.3)	19 (18.1)	31 (18.1)
**Intervention type**				
Pharmacologic therapy	160 (50)	24 (54.5)	54 (51.4)	82 (48)
Radiation therapy	17 (5.3)	1 (2.3)	8 (7.6)	8 (4.7)
Surgical therapy	47 (14.7)	7 (15.9)	13 (12.4)	27 (15.8)
Behavioral/psychological therapy	7 (2.2)	1 (2.3)	0 (0)	6 (3.5)
Herbal/alternative therapy	9 (2.8)	0 (0)	3 (2.9)	6 (3.5)
Other therapy	22 (6.9)	3 (6.8)	6 (5.7)	13 (7.6)
Multiple therapy ^1^	58 (18.1)	8 (18.2)	21 (20)	29 (17)
**External funding**				
None/undisclosed	265 (82.8)	35 (79.5)	92 (87.6)	138 (90.7)
Any	55 (17.2)	9 (20.5)	13 (12.4)	33 (19.5)
Pharmaceutical	34 (10.6)	5 (11.4)	10 (9.5)	19 (11.1)
Non-pharmaceutical	14 (4.4)	2 (4.5)	2 (1.9)	10 (5.8)
Both	7 (2.2)	2 (4.5)	1 (1)	4 (2.3)

^1^ Multiple therapy refers to the use of more than one type of intervention (e.g., combined pharmacologic therapy and radiation therapy, or surgical therapy and radiation therapy) in any given therapeutic cancer trial.

## Data Availability

Data presented in this study are based on the analysis of publicly available data from PubMed and are available on request from the corresponding author.

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
