# Peer review of "Arab Countries and Oncology Clinical Trials: A Bibliometric Analysis"

_cancers, 2023, doi:10.3390/cancers15184428_

Round 1

Reviewer 1 Report

Reviewer Report: Contribution of Arab Countries to Oncology Clinical Trials: A Bibliometric Analysis

General comments

The paper under review performed a bibliometric analysis of publications on cancer research conducted in Arab Countries. The authors did a great job in identifying, collecting, and summarizing relevant studies. I believe the work performed is solid. The paper is also well-written, in general. However, the analysis level seems a bit too shallow, as it does not go beyond some very basic summary exercises. Overall, the paper lacks some analyses that can yield memorable take-away messages.

Specific comments

1. The major theme: The title is called “Contribution of Arab Countries to Oncology Clinical Trials,” but the analysis does not provide a real assessment of the “contribution” of the summarized publications. At first, I thought after summarizing the trends in the literature, the study would perform some analysis on the “impact” of Arab Countries-related cancer research. However, this did not happen. Therefore, I would strongly recommend the authors dig a little deeper into the “contribution” of the summarized publications.   

2. One way to help better assess “contributions” is to see whether cancer studies done in Arab Countries have become proportionally more in the cancer research literature. And did their average impact factor become higher than studies done in other regions?

3. Some other metrics could be considered: how many of these summarized studies were published in top journals? For example: Lancet, Nature, Science, Cell, etc.

4. The nicest part of the current version of the paper is the discussion on the problems identified through your bibliometric analysis. But I think it would be more meaningful to discuss “What should be done next?” Again, since you talk about the “contribution” of this Arab Countries-related literature, what can be potential contributions in the future?

5. Figure 1: Studies done in Egypt account for the vast majority of studies summarized (Why is Egypt so unique in this picture? This may be something to explore more). So, it might be interesting to make some Egypt-non-Egypt comparisons to enrich your findings.

More minor points

1. It will be nice to mention briefly the key parameters of your analysis in the Introduction section to better inform the reader.

2. It would be informative to provide some descriptions of what the “clinical trial studies” you summarized are about. New drugs? New treatments? New therapies? In fact, this line of inquiry might be something to explore.

3. Some studies are of very small sample sizes (e.g., N = 9). You might want to drop some of such studies. 

Reviewer 2 Report

Dear Authors, I read with pleasure your work regarding the state of clinical trials in Arab Countries. Nevertheless, I think that your results should be put more deeply in the context. 

1 - As instance, when you affirm that clinical trials conducted is Arab countries are less likely to be published in journals with high impact factor, this may be linked to the monocentric nature of some trials instead of a geographical disparity. It is comprehensible that monocentric or anyway trials conducted only in one country have less impact than multicentric and international studies. I suggest to add a section discussing this point reporting evidence arising from similar analyses from in countries. 

2- I would remove some auto-citations 

Round 2

Reviewer 1 Report

In response to many of my previous comments, the authors simply did some textual revisions (including putting my comments in the Discussion section as limitations), while my suggestions were to perform some more solid analysis (including getting more "data" from the literature). The revisions are thus largely unsatisfactory. Keeping mentioning my suggestions are "out of the scope of our planned study" only serves to reduce the scientific value of your study, as it narrows down the potential contribution of your study. Simply put, seeing the revisions, I am convinced that my previous "concerns" were really "problems." 
